# Cross-Compatibility in Interspecific Hybridization of Different *Curcuma* Accessions

**DOI:** 10.3390/plants12101961

**Published:** 2023-05-11

**Authors:** Yuanjun Ye, Yiwei Zhou, Jianjun Tan, Genfa Zhu, Jinmei Liu, Yechun Xu

**Affiliations:** 1Guangdong Provincial Key Lab of Ornamental Plant Germplasm Innovation and Utilization, Environmental Horticulture Research Institute, Guangdong Academy of Agricultural Sciences, Guangzhou 510640, China; yeyuanjun@gdaas.cn (Y.Y.);; 2Key Laboratory of Urban Agriculture in South China, Ministry of Agriculture, Guangzhou 510640, China

**Keywords:** cross-compatibility, fertility evaluation, interspecific hybridization, urban landscape, Zingiberaceae

## Abstract

*Curcuma* is extensively cultivated as a medicinal and ornamental plant in tropical and subtropical regions. Due to the bright bract color, distinctive inflorescence and long blooming period, it has become a new favorite in terms of the urban landscape, potted flowers and cut flowers. However, little research on breeding new cultivars using traditional plant breeding methods is available on the genus *Curcuma*. In the present study, pollen viability and stigma receptivity evaluation were performed, and the genetic relationship of 38 *Curcuma* accessions was evaluated, then 5 *C. alismatifolia* Gagnep. (Ca), 2 *C. hybrid* (Ch), 2 *C. sparganiifolia* Gagnep. cultivars and 4 *Curcuma* native species were selected as parents for subsequent interspecific cross-breeding. A total of 132 reciprocal crosses were carried out for interspecific hybridization, including 70 obverse and 62 inverse crosses. Obvious discrepancies among fruit-setting rates were manifested in different combinations and in reciprocal crosses. Results showed that the highest fruit-setting rate (87.5%) was observed in the Ca combinations. There were 87 combinations with a fruit-setting rate of 0%, which meant nearly 65.9% was incompatible. We concluded that *C. alismatifolia* ‘Siam Shadow’ (Ch34) was suitable as a male parent and *C. petiolata* Roxb. (Cpet) was suitable as a female parent to improve the fruit-setting rates. The maximum number of seeds per fruit (45.4) was obtained when *C. alismatifolia* ‘Chiang Mai Pink’ (Ca01) was used as a female parent followed by *C. attenuata* Wall. ex Baker (Catt) (42.8) and *C. alismatifolia* ‘Splash’ (Ca63) (39.6) as male parents. The highest germination rate was observed for the Ca group followed by Catt and *C. sparganiifolia* ‘Maetang Sunrise’ (Csms). The germination rates of Ca accessions ranged from 58.2% (*C. alismatifolia* ‘Siam Scarlet’ (Ca06) as a male parent) to 89.3% (*C. alismatifolia* ‘Sitone’ (Ca10) as a male parent) with an average value of 74.0%. Based on the results of hybrid identification, all the individuals from the four combinations exhibited paternal-specific bands, indicating that the true hybrid rates of crossings were 100%. Our results would facilitate the interspecific hybridization and introduction of genetic variation from wild species into the cultivars in *Curcuma* in the future, which could be helpful in realizing the sustainable application in urban green areas.

## 1. Introduction

The genus *Curcuma* (family Zingiberaceae) comprises more than 110 native species and is distributed in tropical and subtropical regions [1,2,3]. The leaves and rhizomes of *Curcuma* accumulate considerable amounts of active components such as phenols, flavonoids and saturated fatty acids [4,5,6,7]. The flowers have a wide range of medicinal and ornamental uses [8,9,10]. *C. alismatifolia* Gagnep. (2n = 32), commonly known as Siam tulip or Patumma, is a perennial bulbous flowering plant native to Thailand and widely grown in China [11,12] (Figure 1). Patumma has a long flowering duration and blooms continuously from June to October in Guangdong province [13]. With the policies of beautiful countryside and high-quality development of agriculture, this special flower has promising applications for blossom oceans, urban park landscapes, flower-bed landscapes and potted flowers [14]. Moreover, the vase life of Patumma is as long as 15 d. Since the demand for the flower on the market is scarce in summer and autumn, Patumma has become a new favorite cut flower [15]. Other native species of *Curcuma* such as *C. petiolata* Roxb., *C. roscoeana* Wall. and *C. yunnanensis* N. Liu & S. J. Chen exhibit high plant architecture, long inflorescences and bright bracts, which are also known as novel and splendid urban plants [16]. Exploring the potential uses of these valuable resources for improving the ornamental characteristics such as flowering time, bract color and plant architecture is becoming a key question to be addressed in the breeding of *Curcuma*. So far, most *Curcuma* cultivars have been cultivated by foreign breeders [17]. An increasing number of cultivars have been bred for their excellent ornamental characteristics and resistance. Interspecific cross-breeding among *Curcuma* germplasm resources is a new, rising topic, but relevant research is still scarce in China. Because of the popularization of intellectual property protection, it is essential to develop such cultivars for different types of landscape applications.

Unlike many plants, the genus *Curcuma* lacks a genetic transformation system, which makes it difficult to achieve feasible goals in molecular breeding [18,19]. Using traditional breeding methods such as conventional hybridization, radiation mutagenesis and chemical mutagenesis, many rapid improvements on ornamental traits have been achieved [20,21]. Abdullah et al. reported that 25 Gy of gamma irradiation was an effective dose of radiation [22]. After radiation mutagenesis, the bulb germination time, plant height, number of leaves and plant architecture were significantly improved. However, mutagenesis is random rather than directed breeding, and the induced mutations are frequently unstable, where the mutant traits disappear after several generations of reproduction [23,24]. Interspecific hybridization is a common method for breeding new cultivars [25,26]. With different hybrid combinations, new cultivars with stable excellent traits can be quickly obtained [27]. This method has been successfully applied in breeding programs of eggplant, loquat, azalea, tulip and other plants [28,29,30,31]. However, few reports have described the development of interspecific hybridization breeding in the genus *Curcuma*.

The genus *Curcuma* has a complex genetic background with different chromosome numbers, viz., *C. alismatifolia* (2n = 32), *C. thorelii* Gagnep. (2n = 34, 36), *C. parviflora* Wall. (2n = 24, 28, 34, 36, 56), *C. petiolata* (2n = 42, 64) and *C. roscoeana* (2n = 42) [32,33]. Many studies have been carried out to clarify the genetic relationships among the resources of *Curcuma* [34,35]. Taheri et al. used eight SSR markers to analyze the genetic diversity of five varieties and 25 lines of Patumma. Those five varieties were finally divided into two groups [36]. Záveská et al. analyzed the genetic evolution of 19 species in the genus *Curcuma* and explored the evolution of the genus in distant hybridization [37]. Many varieties of *Curcuma* that are bred by artificial crosses have many advantages in bract color and growth characteristics. However, their fertility is too poor to be used as parents for the next steps of breeding, which poses a substantial obstacle to interspecific hybridization. Ketmaro et al. measured the pollen viability of the hybrids *C. sparganiifolia* × *C. parviflora* and found the pollen was almost sterile, but the vigor was improved after colchicine treatment [38]. Saensouk et al. reported the pollen scanning information of 13 germplasm resources of *Curcuma*, which provided a good view for cross-breeding [39]. Yu et al. investigated the pollen storage and viability of 14 *C. alismatifolia* cultivars to provide a reference for *C. alismatifolia* and related species breeding programs [40]. Taking into account that precious genotypes are integral parts of breeding practice, it is pressing to set up a large number of hybrid combinations. The success rate of crossing may be enhanced by selecting parents with close genetic relationships and good fertility.

In this study, through genetic analysis and fertility assessment of *Curcuma* germplasm resources, 13 appropriate parents, including 5 *C. alismatifolia* cultivars (Ca), 2 *C. hybrid* cultivars (Ch), 2 *C. sparganiifolia* cultivars (Cspp and Csms), *C. petiolata* (Cpet), *C. attenuata* Wall. Ex Baker (Catt), *C. thorelii* (Ctho) and *C. yunnanensis* (Cyun), were selected for subsequent interspecific cross-breeding (Appendix A). The aim of the study was to explore the cross-compatibility of interspecific hybridizations between *Curcuma* cultivars and wild species. The fruit-setting rate, seed number and germination rate were calculated in order to obtain numerous seedlings. The information on interspecific crossability is important in developing a comprehensive breeding strategy. The breeding materials employed in this program will help in further improvement of *Curcuma* particularly in bract color and inflorescence height breeding. Our results would facilitate the interspecific hybridization and introduction of genetic variation from wild species into the cultivars in *Curcuma* in the future*,* which could be helpful to realize the sustainable application in urban green areas.

## 2. Results

### 2.1. Fertility Evaluation of Curcuma Germplasm Resources

Based on the fertility evaluations of two methods, there were significant differences among species or cultivars (Figure 2A–F and Figure 3). In general, the *C. alismatifolia* (Ca) samples exhibited the strongest pollen viability. The mean value of fertile pollen and sterile pollen was 71.0% and 11.0%, respectively. The *C. hybrid* (Ch) cultivars showed low pollen viability where the mean fertile pollen accounted for 18.6%, but 59.9% of the pollen was sterile. For the other accessions, *C. petiolata* (Cpet) and *C. sparganiifolia* ‘Pink Pearl’ (Cspp) had excellent fertility, the viable pollen accounted for more than 50%. While the viable pollen of *C. sichuanensis* X. X. Chen (Csic), *C. kwangsiensis* S. G. Lee & C. F. Liang (Ckwa) and *C. yunnanensis* (Cyun) accounted for less than 10% and the inactive pollen was more than 60%. In terms of different groups, Ca accessions showed the best pollen viability, and fertile pollen was 91.3% in *C. alismatifolia ‘*Siam Scarlet*’* (Ca06) and 89.6% in *C. alismatifolia ‘*Holland Red*’* (Ca03). In the Ch group, the best pollen viability was recorded in *C. hybrid ‘*Solo*’* (Ch35) (47.3%) and the weakest pollen viability was found in *C. hybrid ‘*Linglongfen*’* (Ch16), *C. hybrid ‘*Ban Rai Red*’* (Ch47) and *C. hybrid ‘*DT602’ (Ch54) (0%).

The Ca accessions had the best stigma receptivity indicated by the bubbling test (Figure 2G–I). All other samples had strong stigma receptivity except *C. alismatifolia ‘*Silk*’* (Ca42), *C. alismatifolia ‘*Silk*’* (Gongfen) and *C. alismatifolia ‘*Siam Solar*’* (Ca60). The stigma receptivity of Ch cultivars was less than Ca cultivars, whereas Ch47, Ch54 and *C. hybrid ‘*Hong Bai*’* (Ch59) had lower stigma receptivity with almost no bubbles on the stigma. For the other *Curcuma* accessions, Cpet, Ctho, Cspp and *C. sparganiifolia* ‘Maetang Sunrise’ (Csms) showed strong stigma receptivity, while Csic, *C. nankunshanensis* N. Liu, X. B. Ye & Juan Chen (Cnan1), and *C. kwangsiensis* var. *nanlingenesis* (Cnan2) exhibited low stigma receptivity (Table 1).

### 2.2. Genetic Relationship Analysis

A total of 18 EST-SSR markers were selected to screen the polymorphism of 38 *Curcuma* accessions (Appendix A). A total of 173 polymorphic loci were amplified with 9.611 polymorphic loci for each marker. The polymorphic information content (PIC) was the important index for evaluating the polymorphism of each locus where it ranged from 0.628 to 0.888 with an average value of 0.768, indicating that the highly polymorphic EST-SSR markers could be used for genetic analysis of *Curcuma* germplasm resources. According to the clustering results, the 38 germplasms were divided into three groups (Appendix A). The first group contained 17 germplasms, including 4 Ch and 13 Ca cultivars. The common characteristics of these germplasms were that they have long narrow leaves and good fertility. The second group contained 10 specimens, including 2 Cspa and 8 Ch accessions. The common characteristics of them were that they have large leaves and poor fertility. The third group contained 11 samples, of which 9 samples were native species. Based on the results of pollen viability and stigma receptivity, five Ca, two Ch, two Cspa cultivars and four *Curcuma* species were selected as parents for subsequent interspecific cross-breeding.

### 2.3. Fruit-Setting Rates of Different Hybrid Groups

A total of 132 reciprocal crosses were carried out for interspecific hybridization, including 70 obverse and 62 inverse crosses (Appendix A). Based on the results of fruit-setting rates of different hybrid groups, the highest fruit-setting rate (87.5%) was observed in the Ca combinations (Appendix A). There were 87 combinations with a fruit-setting rate of 0%, accounting for 65.9% of the total number of cross combinations. Among the different hybrid groups, the combinations between *C. alismatifolia* interspecies (Ca × Ca) exhibited the highest fruit-setting rate. Besides, the Ca combinations with Ctho and Cpet exhibited low fruit-setting rates. Compared to Ca and other *Curcuma* species, Ch and Cspa preferred to make successful hybridizations with Ctho and Cspa cultivars. For the four native species, Ctho was regarded as the best parent with moderate fruit-setting rates when crossing with Ca, Ch and Cspa accessions. However, all the hybridizations failed when Cyun crossed with the other species or cultivars.

Obvious discrepancies among fruit-setting rates were manifested in different combinations and in reciprocal crosses (Figure 4). Overall, higher fruit-setting rates were observed from obverse crosses for Ca06, Ca09, Ch29, Csms and Cpet, while the fruit-setting rates from inverse crosses were higher than those from obverse crosses for Ca01, Ca10, Ca63, Ch34, Ctho, Cspp and Catt. Results indicated that most accessions could produce fruits as both parents, but there existed some exceptions. For the cross combinations Ca01 × Ch34, Ca01 × Ctho, Ca09 × Ch34, Ca09 × Ctho, Ca10 × Ch34 and Ca63 × Ch34, successful hybridizations were made only in obverse crosses, suggesting that Ch34 was suitable as a male parent for interspecific crossing. For the cross combinations Ca01 × Cpet, Ca06 × Cpet, Ca09 × Cpet, Ca63 × Csms and Ch34 × Ctho, fruits were produced only in inverse crosses, indicating that it was better to choose Cpet as a female parent to improve the fruit-setting rates.

### 2.4. Number of Seeds per Fruit and Germination Rates

The number of seeds per fruit and the germination rates of different hybrid combinations are shown in Figure 5 and Figure 6, respectively. The maximum number of seeds per fruit (45.4) was obtained when Ca01 was used as a female parent followed by Catt (42.8) and Ca63 (39.6) as male parents. There existed significant differences in the number of seeds per fruit between obverse cross and inverse cross. As for Ca01, Ca06 and Csms, a greater number of seeds per fruit were observed in obverse crosses than those in inverse crosses. However, contrasting results were observed for Ca63, Ch29, Ch34, Ctho, Catt and Cpet. The degree of germination rates varied with different hybrid groups. The highest germination rate was observed for the Ca group, followed by Catt and Csms. The germination rates of Ca accessions ranged from 58.2% (Ca06 as a male parent) to 89.3% (Ca10 as a male parent), with an average value of 74.0%. As for Ctho, Cspp and Cpet, higher germination rates and a greater number of seeds per fruit were observed in obverse crosses than those in inverse crosses. For the cross combinations Ca01 × Ch34, Cpet × C01, Ctho × Ca06, Ctho × Ch34 and Cspp × Ctho, the seeds were obtained but did not sprout, indicating the seed development problem resulted from the cross compatibilities of some interspecific hybrids.

### 2.5. Hybrid Identification

Six plants from each of Ca01 × Ca10, Ca06 × Ctho, Ctho × Csms and Catt × Cpet crosses were randomly selected for hybrid identification. The EST-SSR markers with clear bands and strong polymorphism were employed for analysis of the genetic polymorphism of the parents (Appendix A). The results showed that all individuals from four combinations exhibited paternal-specific bands, indicating that the true hybrid rates of crossings were 100% (Table 2).

## 3. Discussion

Interspecies or intergeneric hybridization is one of the main methods for breeding new cultivars in plants [41]. Reports on the fertility of interspecies hybrids of *Curcuma* are scarce, although it is an excellent ornamental and edible medicinal plant with promising market potential. Until now, a large number of excellent cultivars have also been obtained through interspecific hybridization [11,42]. Parent fertility is one of the main factors affecting the seed-setting rate of interspecific hybridization [43,44]. It is easy to overcome the incompatibility of hybridization by using species or cultivars with strong stigma receptivity as female parents. Choosing varieties with high pollen vigor or germination rates as male parents can substantially increase the fruit-setting rate [45]. In this study, the fertility and stigma receptivity of Ca cultivars was generally higher than other *Curcuma* accessions, which could be used by both male and female parents. For some special parents such as Ch34, Ctho and Cpet, it is necessary to comprehensively compare the pollen viability and stigma receptivity of the parents to optimize the cross. Meanwhile, the ploidy level and the chromosome number of the parents may also affect the compatibility of the parents [46]. If the parents have the same ploidy and chromosome number, it would be easier to obtain hybrids. Previous reports found that most Ca cultivars and *Curcuma* native species were diploid, and the chromosome numbers were widely inconsistent [32,33]. Catt had the most chromosome numbers (2n = 84), followed by Cpet (2n = 42, 64), Ctho (2n = 34, 36) and Ca (2n = 32). Most hybridizations between Ca/Ch and *Curcuma* native species were incompatible, which may have been caused by different chromosome numbers. It was concluded that the poor fertility of Ch accessions might have been due to the unsuccessful pairing of chromosomes during meiosis, resulting from different numbers of chromatids and reproductive disorders. We confirmed the fertility of Ch cultivars and provided a view for subsequent interspecific hybridization. However, there were no reports on the chromosome numbers and karyotype analysis of different Ch cultivars. Whether the chromosome numbers are significantly related to fertility or not is still unknown, and the question should be further addressed by studying chromosome compression and karyotype analysis.

Due to the distant genetic relationship of germplasm resources, hybrid incompatibility exists for most *Curcuma* hybridizations. In this study, 18 EST-SSR markers with a mean value of 0.768 were used to assess the genetic relationships among 38 *Curcuma* accessions. Interestingly, the Ca and Ch cultivars and native species were divided together with a few exceptions, which were consistent with their origins and previous findings [12,17]. Combining with the results of the fertility evaluation, the suitable parents from different groups were selected in order to improve the success rates of crossing. A total of 132 interspecific crosses were carried out, which suggested the fruit-setting rates of the hybrids with close parent relationships were significantly higher than those with distant relationships. The four native species had very low success rates of crossing with Ca and Ch accessions, indicating that these species were less genetically related. Especially in Cyun, no fruits were observed for all the combinations. In terms of the low fruit-setting rates, we speculated Ch cultivars have been crossed for multiple generations, which hence may harbor parts of the genome of the native species and Ca cultivars. The Ch cultivars would be suitable as intermediate materials for cross-breeding through introgression. However, due to the limited native species and interspecific hybrid combinations, it is necessary to investigate more into this topic in the future to further verify the influence of genetic relationships on cross-compatibility.

Different parental combinations of obverse and inverse crosses have a great influence on the success rates of crossing. It was found that in the distant hybridization of loquat (*Eriobotrya japonica* Lindl.), the obverse crosses of loquat with its related genera were almost incompatible [29]. When different species were used as male or female parents, the fruit-setting rates were quite different. As a female parent, azalea exhibited better compatibility with the same subgenus *Rhododendron* [30]. However, when the *Rhododendron ovatum* (Lindl.) Planch ex Maxim. was used as the female parent, it failed to cross with *Rhododendron ellipticum* Maxim. but had better crossing compatibility with other species of *Rhododendron*. Similar results were found for the interspecific hybridization of the *Rosa hybrida* E. H. L. Krause, *Iris sibirica* L. and *Prunus* [47,48,49]. For the Ca hybrid group, there were good seed-setting rates in both of the obverse and inverse crosses. Though Ch34 had low stigma receptivity, all the cross combinations failed when used as a female parent. For the native species, except Cyun, there existed significant differences in reciprocal crosses in the fruit-setting rate. The above results preliminarily indicated that the combination of obverse and inverse crosses had a certain effect on the success rates of crosses in *Curcuma*, but no obvious unidirectional hybridizations were observed. By setting up reasonable combinations of obverse and inverse crosses and choosing appropriate parents, there would be a chance to obtain superior hybrids [28]. Although some hybrid combinations showed lower success rates of crossing, there was no obvious fertilization disorder, suggesting that the number of hybrids could be increased by increasing pollination.

When Ch, Ctho and Cspa were used as parents, the number of seeds per fruit and the seed germination rates were lower than other hybrid combinations. Although these combinations produced fruits, the number of seeds per fruit was very little and there were fewer plump seeds. Some fruits of these crosses were even dropped after pollination, which indicated that the abnormal embryo development and incompatibility of pre-zygotic and post-zygotic stages might occur in the process of interspecific hybridization [50]. As for the low seed germination rates, it might have been caused by an interruption of seed development in the zygote stage or abnormalities in the endosperm that led to impaired nutrient supply for the embryo during germination [51]. In order to promote ovary enlargement and increase the fruit-setting rate, delaying pollination before the pollen is fully mature or applying pollen culture solution should be considered [52].

In summary, factors affecting cross-compatibility are complex and diverse. In addition to these objective reasons, environmental conditions such as temperature, humidity and light may also affect hybrid compatibility. To fully understand the causes of cross-compatibility of the genus *Curcuma*, in-depth studies on the physiological and molecular aspects are needed. It is easy to obtain hybrids by interspecific hybridization in which parents are closely related, but it may be difficult to acquire excellent traits. Therefore, it is necessary to try multi-generation backcrossing, selfing and distant hybridization, which will provide more probabilities to create new *Curcuma* germplasms and support breakthroughs in *Curcuma* breeding.

## 4. Materials and Methods

### 4.1. Plant Materials

A total of 38 *Curcuma* accessions were collected from the resource garden of the Environmental Horticulture Research Institute, Guangdong Academy of Agricultural Sciences, Guangzhou, China, which included 9 *Curcuma* native species, 13 *C. alismatifolia*, 2 *C. sparganiifolia* cultivars and 14 *C. hybrid* cultivars (Appendix A).

### 4.2. Pollen Germinability Test

Fresh pollen samples were collected on a flowering day (9–11 a.m.) and placed onto glass slides. The pollen was then killed by heating to a high temperature and used as a control. Fresh pollen samples on glass slides were treated with 1–2 drops of 0.5% 2,3,5-triphenyl tetrazolium chloride (TTC) staining method, mixed evenly, covered with cover glasses, and incubated at 37 °C. The pollen vitality depended on the staining color, which indicated that the dark stain colors were categorized as strong vitality, light colors as medium or low vitality and unstained as no vitality. Meanwhile, the pollen viability above was estimated by in vitro germination on a culture medium referring to the method of Xing et al. [31]. Three flowers were used for each sample, and pollen from three microscope fields was counted for each combination. The total and germinating pollen grains were counted for each field. The pollen staining and germination were observed under a microscope (objective and eyepiece lenses 10×, respectively) (Olympus, CH-20i, Tokyo, Japan).

### 4.3. Evaluation of Stigma Receptivity

The stigmas were collected on a flowering day (9–11 a.m.), applied onto grooved glass slides and immersed into benzidine-H_2_O_2_ reaction solution (1% benzidine: 3% H_2_O_2_:H_2_O, 4:11:22) [51]. The intensity of effervescence on stigmas was observed with a magnifier after 6 min. Color change and strong bubbling in the reaction solution indicated strong receptivity of the stigma. Less bubbling indicated lower receptivity, whereas no bubbling indicated no receptivity. Three replicates for each test were used.

### 4.4. Interspecific Hybridization

To select suitable parents to set up hybrid combinations, EST-SSR molecular markers were used to analyze the genetic diversity of 38 *Curcuma* accessions. The primer information, amplification procedure and genetic diversity analysis referred to the research of Ye et al. (Appendix A) [17]. According to the results of fertility assessment and cluster analysis, five *C. alismatifolia*, two *C. hybrid*, two *C. sparganiifolia* cultivars and four *Curcuma* native species were selected as parents. The interspecific hybridization was conducted in all possible combinations with reciprocal crosses, including 70 obverse and 62 inverse crosses. Wilting florets and fertile bracts around the florets were removed before artificial pollination. Pollen from each male parent was collected and applied to the stigma of the female plant at 9–11 o’clock on the flowering day. The pollinated florets were bagged and labeled. Three days after pollination, the bags were removed, and pod set rate was determined after one month. The seeds in each fruit were harvested and counted after ripening. They were sown in March of the next year, and the germination rate of each combination was calculated (Figure 7).

### 4.5. Hybrid Identification

When the seedlings were at the two-leaf stage, four hybrid combinations were randomly selected, and six plants of each combination were chosen for DNA extraction. Their DNA was extracted using a new-type genomic DNA extraction kit (Tiangen Biotechnology Co., Ltd., Beijing, China). The integrity and quantity of DNA were evaluated by 1% agarose gel electrophoresis and a Nanodrop 2000 spectrophotometer (Thermo Fisher Scientific, Waltham, MA, USA), respectively. To perform the PCR amplification, the forward primer of EST-SSR markers was elongated from the M13 primer appended to the 5’-end [17]. PCR amplifications were conducted on 15 µL volumes containing 7.5 µL of 2× *Taq* mix, 0.2 µL of forward M13 primer (1 µM), 1.2 µL of reverse primer (1 µM), 1.2 µL of M13 fluorescent primer (1 µM), 2.5 µL of template DNA (10 ng/µL) and 2.4 µL of ddH_2_O. The thermal cycling program consisted of pre-denaturation at 94 °C for 5 min, 30 cycles at 94 °C for 30 s, 55 °C for 30 s and 72 °C for 1 min, followed by 13 cycles at 94 °C for 30 s, 53 °C for 30 s and 72 °C for 1 min, and a final extension at 72 °C for 10 min. The marker bands in the parents were screened using the polyacrylamide gel electrophoresis, and approximately 0.5 μL of PCR products with four different fluorescent labels (FAM, HEX, ROX and TAMRA) and sizes were pooled and detected using a DNA analyzer. Finally, the hybrids with paternal bands were identified as true hybrids.

## 5. Conclusions

As promising urban plants for different types of landscape applications, it is pressing to breed new *Curcuma* cultivars with excellent ornamental characteristics and resistance. In this research, a total of 132 reciprocal crosses were conducted among 13 *Curcuma* germplasm resources by assessing the fertility and genetic relationship. Significant differences among fruit-setting rates were manifested in different combinations and in reciprocal crosses. Results showed that the highest fruit-setting rate (87.5%) was observed in the Ca combinations. The maximum number of seeds per fruit (45.4) was obtained when Ca01 was used as the female parent followed by Catt (42.8) and Ca63 (39.6) as the male parent. The highest germination rate was observed for the Ca group followed by Catt and Csms. All the seedlings obtained from the crosses were true hybrids through hybrid identification. The present study was performed to explore the crossability behavior of different *Curcuma* accessions and to develop innovative cultivars with desirable traits. Our results would facilitate the interspecific hybridization and introduction of genetic variation from wild species into the cultivars in *Curcuma* in the future.

## Figures and Tables

**Figure 1 plants-12-01961-f001:**
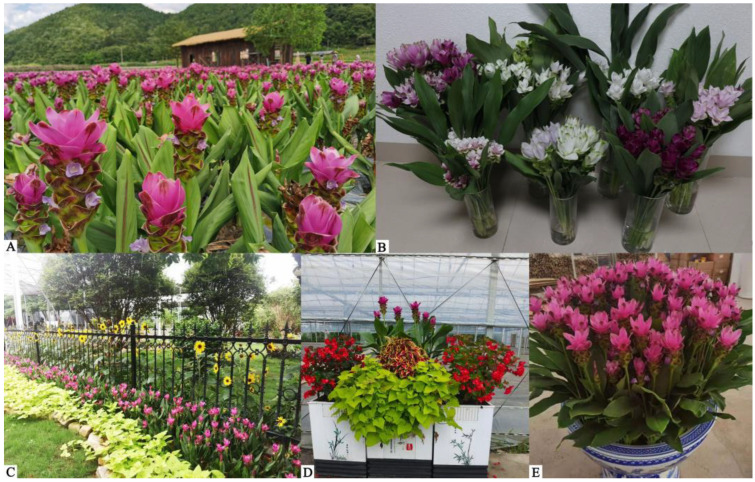
Different types of landscape applications of *C. alismatifolia: (***A**) Blossom ocean of beautiful countryside; (**B**) Fresh cut flower; (**C**) Greening-road or urban park landscape; (**D**) Flower-bed landscape; (**E**) Potted flower.

**Figure 2 plants-12-01961-f002:**
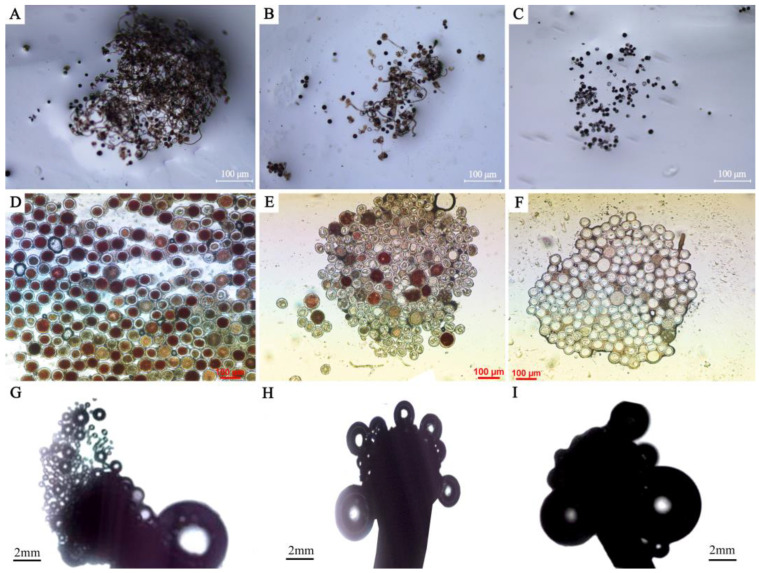
In the fertility evaluation of different *Curcuma* accessions, the selected samples are Ca01, Csms and Ch59 from left to right: (**A**–**C**) Pollen germinability test by in vitro germination on culture medium; (**D**–**F**) Pollen germinability test using TTC staining method; (**G**–**I**) Stigma receptivity evaluation.

**Figure 3 plants-12-01961-f003:**
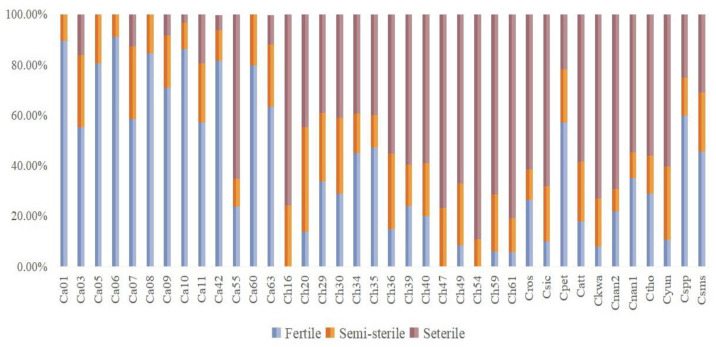
The pollen viability of 38 experimental *Curcuma* accessions.

**Figure 4 plants-12-01961-f004:**
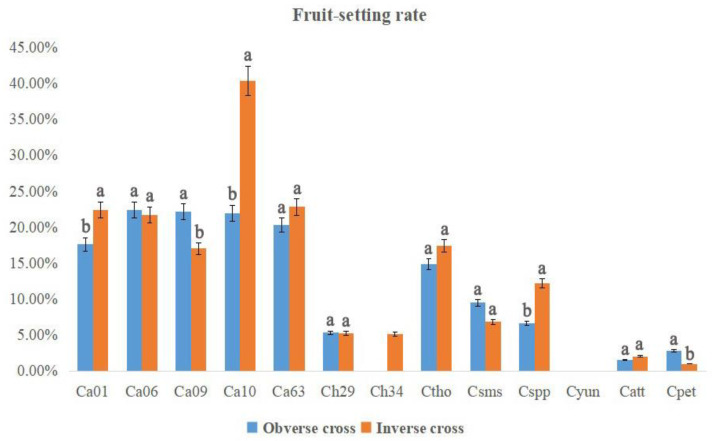
The fruit-setting rates in each cross combination of *Curcuma*. Different lowercase letters represent significant differences (*p* < 0.5).

**Figure 5 plants-12-01961-f005:**
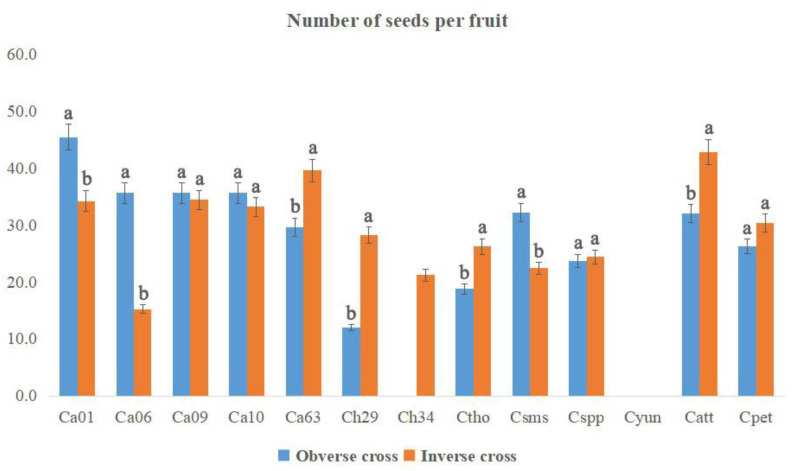
Number of seeds per fruit in each cross combination of *Curcuma*. Different lowercase letters represent significant differences (*p* < 0.5).

**Figure 6 plants-12-01961-f006:**
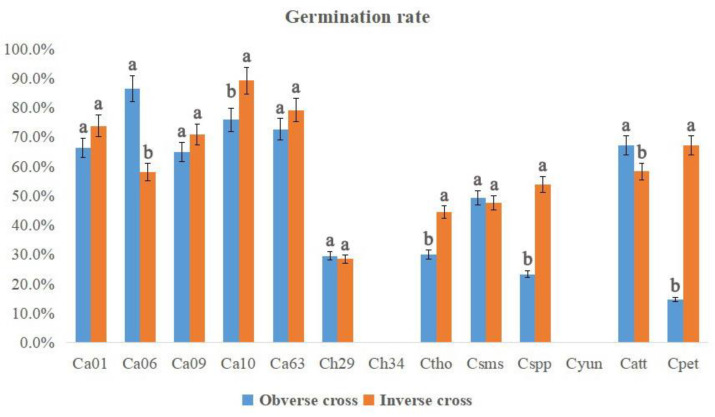
Germination rates in each cross combination of *Curcuma*. Different lowercase letters represent significant differences (*p* < 0.5).

**Figure 7 plants-12-01961-f007:**
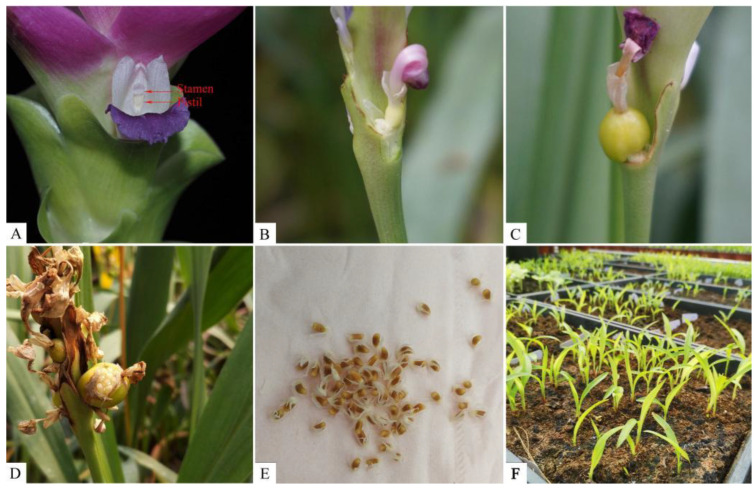
The schematic process of cross-breeding: (**A**) The stamen and pistil of *C. alismatifolia*; (**B**) The pollinated floret after one week; (**C**) The pollinated floret after three weeks; (**D**) The pollinated floret after five weeks; (**E**) The seeds of *C. alismatifolia*; (**F**) The seedlings of *C. alismatifolia*.

**Table 1 plants-12-01961-t001:** The Stigma receptivity evaluation of 38 *Curcuma* accessions.

Accessions	Stigma Receptivity	Accessions	Stigma Receptivity	Accessions	Stigma Receptivity
Ca01	+++	Ch16	+	Ch61	+
Ca03	+++	Ch20	++	Cros	++
Ca05	+++	Ch29	++	Cpet	+++
Ca06	+++	Ch30	++	Csic	+
Ca07	+++	Ch34	+	Ckwa	+
Ca08	+++	Ch35	+	Catt	++
Ca09	+++	Ch36	++	Cnan1	+
Ca10	+++	Ch39	+	Cnan2	+
Ca11	+++	Ch40	++	Ctho	+++
Ca42	++	Ch47	—	Cyun	++
Ca55	++	Ch49	+	Cspp	+++
Ca60	++	Ch54	—	Csms	+++
Ca63	+++	Ch59	—		

Note: +++ means stigmas have high receptivity; ++ means stigmas have medium receptivity; + means stigmas have low receptivity; — means stigmas have no receptivity.

**Table 2 plants-12-01961-t002:** Results of amplification for parents and individuals using EST-SSR markers.

SSR Locus	Parent		F_1_ Individual						Purity%
	Ca01	Ca10	Caa-1	Caa-2	Caa-3	Caa-4	Caa-5	Caa-6	100
JHH10	160/160	160/166	160/166	160/166	166/166	160/166	166/166	166/166	
	Ca06	Ctho	Cat-1	Cat-2	Cat-3	Cat-4	Cat-5	Cat-6	100
JHH2	245/248	248/251	251/251	248/251	245/251	245/251	248/251	251/251	
JHH21	146/155	146/149	146/149	146/149	146/149	149/155	149/155	146/149	
	Ctho	Csms	Cts-1	Cts-2	Cts-3	Cts-4	Cts-5	Cts-6	
JHH10	166/172	169/172	169/172	169/169	169/172	169/172	169/169	166/172	100
JHH2	248/248	248/251	251/251	248/251	248/248	248/251	251/251	248/251	
	Catt	Cpet	Cap-1	Cap-2	Cap-3	Cap-4	Cap-5	Cap-6	100
JHH15	130/133	130/139	130/139	130/139	139/139	139/139	130/139	130/139	

## Data Availability

The data presented in this study are available on request from the corresponding author.

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
