# Peer review of "Cross-Compatibility in Interspecific Hybridization of Different Curcuma Accessions"

_plants, 2023, doi:10.3390/plants12101961_

Round 1
Reviewer 1 Report
The paper is very interesting and worth publishing. The paper deeply explain the problems with hybridization between well known several varieties Curcuma alismatifolia and less known other interesting species. Curcuma is very interesting and valuable as horticultural plant. In spite of using them in landscape also big potential is to use them as cut flowers and rhizome production for commercial cultivation by horticulturist. Breeding of any species is not easy but the presented paper deeply explain the problems with fertility assessment of diffent Curcuma species (pollen viability and stigma receptivity). Also the genetic relationships (using EST-SRR markers) were studied to choose the best parents between Curcuma species and varieties. After pollination, the other parameters as fruit-setting rate, number of seeds per fruit and germination of seeds were assessed. Valuable parts of the paper are also photos (fig. 1, fig 2and fig 5) which are very interesting for readers who don't know well this plant.
Some minor corrections are included in the text.

Reviewer 2 Report
The topic is interesting but the paper needs some improvements.
First of all, in my oppinion the objectives and the scope of this work should be better defined and detailed...
The information regarding plant material (parents, crosses, hybrids) have to be better organized, described and included in the article, not in the supplementary material.
I suggest to use the ”International Code for the Nomenclature for Cultivated Plants” in order to have uniform naming for the plant material. See lines 16, 39, 289. Please, check all the names of the species and hybrids, include the ”authority” name(s).
Please, clarify whether are cultivars or ”varieties” (lines 13, 29, 52, 68, 76, 77, 79, 25, 104, 106, 121, 155...).
Final conclusion (lines 345-346 and lines 94-95!) needs more explanation.
Can you specify the novel tools for breeding that you reccomend?
Reviewer 3 Report
Dear Authors,
The present manuscript addresses compatibility in interspecific hybrids in different accessions of Curcuma.
The experimental design is appropriate and the studied parameters are interesting for the readers. The manuscript needs major revision if the following sections:
Abstract:
Line 17 Please add the full botanical name of the different Curcuma species, e.g. Curcuma sparganifolia Gagnep.
Line 20 Do not use abbreviating with are not expanded in the Abstract.
Keywords: Do not use the same word here as in the title! Also, they need to be in alphabetical order.
Introduction: In line 39 please all Latina names of the plant species need to be in Italic script and correctly botanically written! Correct it in the entire manuscript!
Please formulate the clear goal of this study!
Results
Line 102 Figure S1 needs to be added to the manuscript instead Figure 1. It is very unconvinced to have the first result in the supplement!
Entire section 2.1 Fertility evolution…
Need more clear writing, it is difficult to read! Also, the abbreviations are difficult to follow!
Line 114 Figure 2 is very bad quality, Pane A, B, and C is absolutely invisible, the same the G, H, I very poor quality, not if focus, no magnification is mentioned in all pictures! Here also the histological results are missing!
Line 125 Data here presented here are very subjectively presented, these results need statistical analyses!
Lines 127 and 147 Sections 2.2 and 2.3
Line 322 Why this table is in section M&M?
Please add Tables S1 and S1 to the manuscript.
All Figures and Tables need to be reorganized in this manuscript!
Also, the basic statistic needs to be used for presenting the obtained results.
Discussion
Need to be rewritten, here in this section are data with need to go to Results, special the SSR part! Line 209.
Individual paragraphs of the Results section need to be discussed in depth and shown clearly in the Discussion section which now is chaotic.
Material and methods:
This section needs improvement!
Line 297 specified the model of the Olympus microscope.
Here you are writing that the pink coloration is for pollen staining and vitality test. But in your pictures is hardly visible the pick color …!
Line 300 in vitro test is in the Italic script.
Line 307 wrong citation.
Section 4.5 Hybrid identification
Please write in detail how the DNA was extracted and PCR was performed. A witch DNA analyser has been used. This section needs complete rewriting.
26.4.2023
Needs improvement.
Reviewer 4 Report
The paper entitled ‘Cross compatibility in interspecific hybridization of different Curcuma accessions.’ describe novel tools for the breeding of new varieties of Curcuma, which could help implement sustainable applications in urban green areas.
The authors used genetic analysis and the assessment of the fertility of Curcuma 38 germplasm to select appropriate parents. Many hybrid combinations were carried out to obtain excellent traits of potential new varieties of Curcuma sp.
After reading the text, I have a few comments and suggestions.
Major comments:
1. I have a question, it is related to the statistical study. In the figures shown, it is not seen that any statistical test has been carried out, even though the study has been carried out with replicates; could they include it?
Minor comments:
1. The list of abbreviations used is necessary. It is impossible to follow the text and the entire experiment without it. Even in the abstract, the authors use abbreviations, e.g. Ca, Catt, Cpet…..
2. Line 36. ‘The leaves and bulbs of Curcuma…..’ The bulbous plant is the common name for this kind of plant, but botanically correct in the case of this genus is to use the term rhizome.
3. Line 55. ‘domestic research’ means domestic research for the authors. It is better to specify the country or region.
4. Table 1. Please add the footer with a description of the labels used in columns (+++/ ++/ + / -).
5. The abstract and line 290 - C. Sparganifolia is written in capital letter.
6. Lines 129 and 313. ‘38 Curcuma species’ The term species can not be used because there are also varieties and cultivars.
7. Line 156. What does 'significant differences' mean if the authors did not perform any statistical analysis?
8. Line 204. ‘excellent ornamental and edible medicinal flower’ suggests that flowers are used therapeutically, but it is not true. Better write ‘excellent ornamental and edible medicinal plant’.
Round 2
Reviewer 2 Report
Congratulation for your interesting work!
Reviewer 3 Report
Hello,
thank you for all your corrections and improvement of the manuscript.
Now I ca recommend it for publishing.
8.8.2023